# CoINR: COMPRESSED IMPLICIT NEURAL REPRESENTATIONS

## ABSTRACT

Implicit Neural Representations (INRs) are increasingly recognized as a versatile data modality for representing discretized signals, offering benefits such as infinite query resolution and reduced storage requirements. Existing signal compression approaches for INRs typically employ one of two strategies: 1. direct quantization with entropy coding of the trained INR; 2. deriving a latent code on top of the INR through a learnable transformation. Thus, their performance is heavily dependent on the quantization and entropy coding schemes employed. In this paper, we introduce **CoINR**, an innovative compression algorithm that leverages the patterns in the vector spaces formed by weights of INRs. We compress these vector spaces using a high-dimensional sparse code within a dictionary. Further analysis reveals that the atoms of the dictionary used to generate the sparse code do not need to be learned or transmitted to successfully recover the INR weights. We demonstrate that the proposed approach can be integrated with any existing INR-based signal compression technique. Our results indicate that **CoINR** achieves substantial reductions in storage requirements for INRs across various configurations, outperforming conventional INR-based compression baselines. Furthermore, **CoINR** maintains high-quality decoding across diverse data modalities, including images, occupancy fields, and Neural Radiance Fields.

## 1 INTRODUCTION

Despite the fact that all naturally occurring signals observed by humans are continuous, capturing these signals through digital devices requires their discretization. For example, an image of a mountain is processed and stored in a discretized format. A primary reason for this approach is to conserve storage space; storing signals with high precision in an almost continuous manner would necessitate a substantial amount of storage. Consequently, the digital representation of signals in a discretized form is both practical and essential. For instance, it is estimated that over 400TB of data is created every day (Duarte, 2024). Moreover, humans share their captured signals through various mediums on a daily basis. Therefore, data compression becomes essential for efficient and reliable transmission.

Traditional signal compression techniques often rely on classic signal processing methods and are typically unimodal. For example, JPEG (Wallace, 1992), designed for photographic images, and is unsuitable for audio files. Similarly, audio compression standards like MP3 or AAC (Brandenburg, 1999) are optimized for sound and are not applicable to images. With the advancements of neural networks, researchers have explored compressing signals using neural methods, predominantly through mechanisms based on autoencoders (Alexandre et al., 2018; Cheng et al., 2019; Theis et al., 2022). In these systems, the encoder transforms the signal into a latent vector, which the decoder then uses to reconstruct the original signal. While autoencoder-based methods effectively encode signals into latent vectors, they are generally designed for images or another single modality. Adapting these methods to different data modalities not only requires training on a large corpus of data specific to those modalities but also a specialized autoencoder architecture tailored to handle the data effectively.

In recent years, there has been a significant surge in interest in representing signals through Implicit Neural Representations (INRs). Unlike large models based on autoencoders, INRs typically consist of multi-layer perceptrons (MLPs) equipped with specialized nonlinearities that differ from the con-

ventional nonlinearities used in deep learning. This simplicity and versatility allow INRs to unify signal representations across diverse data modalities. When signals are represented by INRs, they are encoded in the MLP's weights and biases. For instance, in image transmission, instead of using conventional JPEG encoding, the weights and biases of the MLP are transmitted by a transmitter (TX). A receiver (RX) can then feed the coordinates into the MLP and decode the image. The primary advantage of INRs lies in their ability to represent signals with high fidelity while utilizing fewer parameters than parameter-heavy autoencoder-based mechanisms.

Recent advances in INR-based signal compression include COIN (Dupont et al., 2021), COIN++ (Dupont et al., 2022), and INRIC (Strümpler et al., 2022). COIN pioneered the application of INRs for image compression. Building on this, COIN++ and INRIC introduced quantization and entropy coding to improve compression efficiency. Both approaches also focus on enhancing the generalization capabilities of INRs through meta-learning techniques. Additionally, COIN++ incorporates latent modulations discovered via a learnable transformation applied on top of the INR model. However, COIN++ requires transmitting the base INR and the learned transformation apriori, in addition to the latent modulations for signal decoding. None of the existing methods, however, have explored fundamentally compressing the INR by identifying patterns within its parameter space before applying standard techniques such as quantization and entropy coding.

In our work, named **CoINR**, we build upon the observed behaviors of the vector spaces generated by the weights in an INR. We integrate compressed sensing algorithms into the INR-based compression pipeline, proposing a mechanism that obtains a higher-dimensional sparse code for the weight vectors without requiring any learnable transformations. Furthermore, based on the Central Limit Theorem (CLT) (Zhang et al., 2022), we show that the transformation matrix need not be transmitted for successful decoding of weight spaces. This further enhances and simplifies the decoding process. Consequently, **CoINR**, as a fundamental compression technique built on the observations of weights spaces, achieves superior compression and higher decoding quality for each data modality compared to the baselines. Moreover, it can be easily embedded into any INR-based signal compression algorithm.

## 2 RELATED WORKS

### 2.1 IMPLICIT NEURAL REPRESENTATIONS

INRs have recently gained considerable attention in the computer vision community due to their streamlined network architectures and improved performance in various vision tasks compared to traditional, parameter-heavy models (Sitzmann et al., 2020; Saragadam et al., 2023; Hao et al., 2022). This surge in interest followed the advent of Neural Radiance Fields (NeRF) (Mildenhall et al., 2021), which has inspired a plethora of subsequent studies (Zhu et al., 2023; Rabby & Zhang, 2023). Further research has explored the pivotal role of different activation functions in INRs (Sitzmann et al., 2020; Saragadam et al., 2023; Ramasinghe & Lucey, 2022; Tancik et al., 2020). Moreover, INRs have transformed into a unified data modality that integrates various types of visual information into a consistent format. More recent studies have investigated the use of INRs for image classification by transforming standard image formats into INRs and training classifiers directly on the INRs' weights and biases (Shamsian et al., 2024). These innovative approaches have showcased the potential of INRs to significantly reduce the dimensionality and computational complexity typically associated with conventional image processing techniques.

### 2.2 SIGNAL COMPRESSION

Signal compression is crucial for reducing bandwidth needs and saving storage space. With the rise of deep learning, signal compression has evolved into two main approaches: rule-based (traditional) and learning-based methods. Traditional compression methods, such as JPEG for images and MP3 for audio, rely on algorithmic techniques tailored to specific signal types. JPEG minimizes redundancies using the discrete cosine transform (Raid et al., 2014), while MP3 (Brandenburg, 1999) employs a psycho-acoustic model that enhances compression by removing inaudible sounds through auditory masking. On the other hand, deep learning-based techniques use models trained on vast datasets, adapting to a wide range of signals without predefined algorithms. These methods offer flexibility but require different architectures for each data modality, presenting unique challenges.

In this landscape, INRs stand out as a potential universal signal representor. INRs can handle various data types through a unified framework, promising a versatile solution in the realm of signal compression.

### 2.3 COMPRESSED SENSING

Compressed sensing is a field that capitalizes on the inherent sparsity of data to capture information efficiently. In digital imaging, not every pixel is crucial for accurate image reconstruction. Although images appear dense in pixel space, they exhibit considerable redundancy when transformed into different basis functions. This sparsity is exploited by compressed sensing algorithms to reconstruct the original image from fewer sampled data points. These algorithms employ optimization techniques and linear algebra to solve underdetermined systems, revolutionizing data acquisition in areas such as medical imaging and signal processing. Dictionary learning, integral to compressed sensing, seeks sparse representations of data using dictionary elements or atoms that capture the data's intrinsic structure. These atoms are either predefined or adaptively learned. Compressed sensing's versatility is evident in its applications across various domains, such as image and video compression (Zhou & Yang, 2024), medical image encryption (Jiang et al., 2024), and classification tasks (Liu & Fieguth, 2010; Kapoor et al., 2012; Hsu et al., 2009; Hu & Tan, 2018). It also addresses inverse vision problems like image inpainting (Seemakurthy et al., 2020), deblurring (Ma et al., 2013; Hu et al., 2010), and super-resolution (Ayas & Ekinci, 2020). Recent efforts have merged dictionary learning with deep learning to tackle more complex computer vision challenges, including image recognition (Tang et al., 2020), denoising (Zheng et al., 2021), and scene recognition (Liu et al., 2018). These developments underscore compressed sensing's transformative impact on computer vision.

Our work, **CoINR**, is pioneering the application of compressed sensing principles to INRs. By leveraging these principles alongside the structural distributions of INR weights, **CoINR** identifies redundancies in these spaces, resulting in substantial compression improvements.

## 3 METHOD

### 3.1 SIGNAL REPRESENTATION THROUGH INRS

Mathematically, an INR can be defined by a function $G_\theta$, where $\theta$ are the optimizable parameters of the neural network. The input and output dimensions of $G_\theta$ vary for different data modalities. In general, $G_\theta$ acts as a mapping from an $a$-dimensional input coordinate space to a $b$-dimensional output signal space, described mathematically as:

$$G_\theta : \mathbb{R}^a \to \mathbb{R}^b.$$

For instance, for RGB images, $a = 2$ and $b = 3$, while for audio signals, $a = 1$ and $b = 1$. In this architecture, the output of the $i^{\text{th}}$ layer, which feeds into the $(i + 1)^{\text{th}}$ layer, can be expressed as $\sigma(W^{(i)}y^{(i)} + b^{(i)})$. Here, $\sigma$ denotes the activation function, and $y^{(i)}$ represents the output from the preceding layer. Furthermore, the choice of activation function ($\sigma$) plays a critical role in shaping the neural network's ability to model complex functions, as explored in various studies (Sitzmann et al., 2020; Ramasinghe & Lucey, 2022; Saragadam et al., 2023; Tancik et al., 2020).

### 3.2 EXPLORING THE COMPRESSIBILITY

According to compressed sensing theory, most real-world signals display sparsity when transformed into an appropriate domain, meaning they can be accurately represented with fewer measurements than traditionally required. Furthermore, real-world signals can be compressed through a set of basis functions, and the coefficients of these functions are derived by minimizing the reconstruction loss. The core concept of INRs involves encoding signals into the weights and biases of an MLP. This process can be viewed as a classical domain transformation technique where pixel values are reconstructed by feeding the corresponding coordinates through the MLP. Unlike predefined signal transformers like Fourier (Chandrasekharan, 2012) and DCT (Khayam, 2003), the MLP attempts to minimize the reconstruction loss through backpropagation to find the transformation. The learned representation of the signal resides in another domain. Given that real-world signals inherently



(a) Layer 2, and 3 weight distributions when hidden neuron count is 64

(b) Layer 4, and 5 weight distributions when the hidden neuron count is 256

Figure 1: **Weight distribution of INRs follows a Gaussian distribution**: A randomly choosen image from Kodak dataset was fitted through an INR.

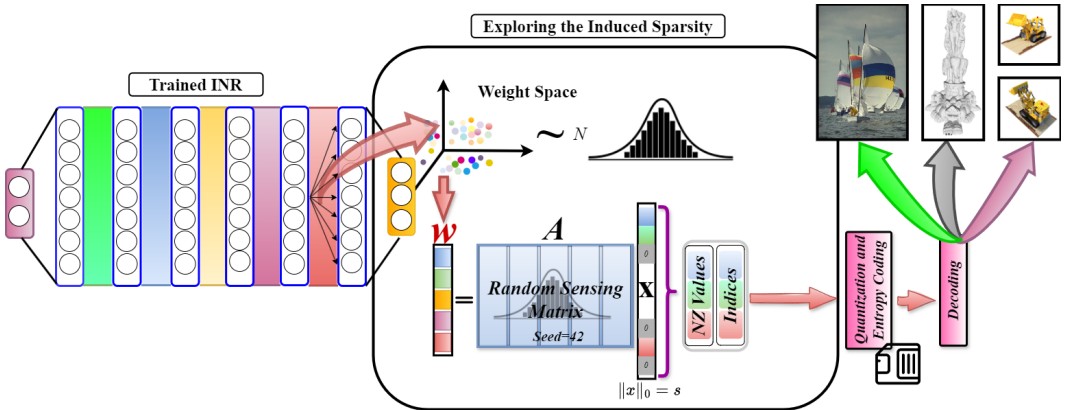

Figure 2: **The proposed `CoINR` compression algorithm** Standard compression techniques for INRs typically involve direct quantization and entropy coding of their weights. However, since natural signals exhibit inherent compressibility in a dictionary, the characteristics that aid in the compressibility of the weight space of an INR are discovered through the Gaussian nature of the weight space. Therefore, `CoINR` employs $L_1$ minimization to identify a higher-dimensional sparse code. Furthermore, based on the weight space observations and the Central Limit Theorem (CLT), we simplify the encoding and decoding process using a random sensing matrix controlled by a seed. Subsequently, only the non-zero (NZ) values and their corresponding indices are quantized and entropy coded.

exhibit sparsity in transformed domains, we hypothesize that this sparsity can be explored within the MLP's weights. If we can identify where this sparse nature is hidden within the weight space, we could achieve further compression on INRs compared to the baselines. However, identifying this sparse representation within the weights is not straightforward. We believe there are two main approaches to achieving a sparser representation, each with its own challenges and considerations.

The first approach involves either promoting or enforcing a specified level of sparsity in the weights during the training of an INR. Promoting sparsity can generally be achieved by incorporating $L1$ regularization on the model parameters, which encourages the model to set as many weights as possible to zero, thereby creating a sparse representation. Enforcing a specified level of sparsity can be achieved through model pruning, where weights deemed insignificant are pruned or eliminated during the training process. Despite our efforts using these techniques, we observed that $L_1$ regularization results in a higher level of sparsity within the weights but fails to showcase a clear pattern of sparsity levels for natural images. When it comes to model pruning, we employed both structured and unstructured pruning of weights. We noted that both methods led to significant performance degradation for certain data modalities, particularly for occupancy fields. Moreover, only a small pruning percentages resulted in satisfactory performance for signal representation. For applications that require a high level of generalization, such as NeRFs, the pruning approach did not generalize well, indicating its limitations in achieving a balance between sparsity and performance.

The second approach seeks to uncover the inherent structures within the weights that aid in INR compression. This involves identifying patterns or regularities that can be exploited to reduce the dimensionality of the representation without sacrificing performance. We examined this from a dimensionality reduction perspective; however, the weight space in reduced dimensions did not reveal clear patterns, even across different natural images. However, we observed that the weight space of an INR follows a normal distribution for every instance of every data modality. Figure 1 shows the weight distribution of hidden layers of an INR when an image is encoded into it. Our observations are further confirmed by analysis done in Sitzmann et al. (2020). This suggests that INRs share a common pattern across different data modalities, showcasing a potential pathway for a fundamental compression.

Given that each weight vector of an INR exhibits Gaussian behavior, we seek a higher-dimensional but sparse equivalent through a dictionary learning-based approach. Let us denote $\mathbf{w} \in \mathbb{R}^{k_1}$ as a hidden weight vector, $\mathbf{A} \in \mathbb{R}^{k_1 \times k_2}$ as a dictionary, and $\mathbf{x} \in \mathbb{R}^{k_2}$ as the corresponding sparse vector. In search of a sparse representation, according to standard compressed sensing, we can write $\mathbf{w} = \mathbf{Ax}$, where $\|\mathbf{x}\|_0 < k_1$. To discover the sparse code $\mathbf{x}$, the best and most efficient choice is $L_1$ minimization, as $L_0$ minimization iterates through all possible combinations and is therefore not efficient. However, the problem arises with the sensing matrix, commonly referred to as the dictionary $\mathbf{A}$. Although we could use either a dictionary learning-based approach for learning basis functions for the dictionary or a deep learning-based learnable transformation, these approaches would be time-consuming. Furthermore, a TX needs to transmit the learned dictionary alongside the obtained sparse codes. Further exploration of the weight space revealed that the dictionary does not need to be learned or even transmitted.

As we have confirmed, the weights are normally distributed. According to the Central Limit Theorem (CLT), a normally distributed random variable can be produced through a finite linear combination of any random variables. In summation form, this can be expressed as: $w_i = \sum_{j=1}^{k_2} A_{ij} x_j$, where $w_i$ is the $i$-th element of the weight vector $\mathbf{w}$, $A_{ij}$ is the element in the $i$-th row and $j$-th column of the sensing matrix $\mathbf{A}$, and $x_j$ is the $j$-th element of the vector $\mathbf{x}$. To satisfy the CLT, the number of terms in the summation, which is $k_2$, should be sufficiently large. Therefore, considering all elements of the weight vector $\mathbf{w}$, this can be compactly written as $\mathbf{w} = \mathbf{Ax}$. As we now understand the structure of the sensing matrix, which is a random matrix, the appropriate coefficients of those random vectors can be learned through the $L_1$ minimization discussed earlier by leveraging dictionary learning algorithms such as matching pursuit or its variants. Therefore, the optimization problem can be written as, $\min \|\mathbf{x}\|_1$ subject to $\mathbf{w} = \mathbf{Ax}$. For convenience, let us denote $\|\mathbf{x}\|_0$ as $s$. A further constraint to the above optimization procedure is that when the sparse code $\mathbf{x}$ is found, we need to store not only its non-zero elements but also the corresponding indices. Therefore, the above $L_1$ minimization is solved with $2s < k_1$. We do not apply our compression algorithm to the biases on the INR as the size of bias vectors is very small compared to those of the weight matrices.

Instead of saving $k_1$ floating-point numbers for $\mathbf{w}$, we now only need to save $2s$ elements: $s$ elements are floating-point numbers representing the non-zero values in the sparse code, and the remaining $s$ elements are integers that give the indices of those non-zero values. The indices can often be represented with 16-bit precision, unlike the non-zero values in the sparse code, which require 32-bit floating-point precision. At the RX end, $\mathbf{x}$ must be converted back to $\mathbf{w}$. This requires the sensing matrix $\mathbf{A}$, which is random and must be controlled by a seed to reproduce the exact $\mathbf{w}$ using $\mathbf{w} = \mathbf{Ax}$. Thus, the receiver only needs $\mathbf{x}$ to obtain $\mathbf{w}$.

This process can be viewed as a method of uncovering the inherent sparsity within natural signals, as represented through the weight space of INRs. As we hypothesized, the ability to condense natural signals into a dictionary hinges on identifying specific patterns encoded within the weights of INRs. Once the non-zero elements of the sparse vector are pinpointed, the resulting procedure is virtually the same across different INR-based baselines. Our method fundamentally achieves compression by delving into the weight spaces to uncover patterns, a step not typically taken by existing baselines. A summarization of **CoINR** is illustrated in figure 2. As can be seen from figure 2, **CoINR** is only dependent on the weights of the INR and is applied prior to any quantization or entropy coding schemes. Therefore, **CoINR** can be applied to any existing INR compression baselines to improve their compressibility.

### 3.3 HOW MUCH FUNDAMENTAL COMPRESSION DOES **CoINR** ACHIEVE COMPARED TO THE BASELINES?

#### 3.3.1 STANDARD INRS

Consider an INR with $l$ hidden layers, yielding $l + 2$ total layers. For simplicity, assume $k$ neurons per hidden layer. If the input dimension is $a$ and the output dimension is $b$, the total number of weight parameters is given by $\mathcal{T}_s = a \times k + l \times k^2 + b \times k$. However, **CoINR** modifies this structure by reducing the parameters from $\mathcal{T}_s$ in the original network to $\mathcal{T}_{s\text{CoINR}} = a \times 2s + k \times l \times 2s + b \times 2s$, where $s \ll k$. Additionally, unlike COIN++, **CoINR** does not require transmitting any additional data to recover the original INR weights.

#### 3.3.2 TINY INRS

Let us define an INR as "tiny" if the number of neurons in a hidden layer, denoted by $k$, is less than 50. In such cases, we aim to achieve a sparse representation where $2s < k$ and $\|x\|_0 = s$. However, achieving a sparse representation that satisfies $2s < k$ is often extremely challenging and typically does not result in effective compression. To overcome this, we exploit the fact that the weight matrix connecting the $i^{\text{th}}$ layer to the $(i + 1)^{\text{th}}$ layer is of dimensions $k \times k$. By vectorizing this weight matrix, we obtain a vector of dimension $k^2 \times 1$. Given that $k^2$ is significantly larger than $k$, we can apply our **CoINR** procedure directly to the flattened weight matrix. This strategy leads to a sparser representation, thereby enhancing compression efficiency for tiny INRs.

#### 3.3.3 COIN++

In the COIN++ framework, modulation parameters are stored instead of traditional weights and biases, under the assumption that the base network parameters can be transmitted beforehand. For $n$ test images, each segmented into $m$ patches with a latent dimension of size $d$, COIN++ necessitates the transmission of $m \times d$ parameters for reconstructing each image. As the base network in COIN++ conforms to a standard INR structure, it is amenable to further compression via the **CoINR** technique. By implementing **CoINR** principles on the modulations in COIN++, the parameter transmission requirement per image can be reduced from $m \times d$ to just $2s \times d$, where $s \ll m$. As the size of each test image and the number of images in the test dataset grow, COIN++ would typically require the transmission of numerous parameters. However, by leveraging **CoINR**, both the modulations and the base network can be significantly compressed, achieving enhanced compression.

### 3.4 QUANTIZATION AND ENTROPY CODING

After an INR is trained, its parameters are not immediately saved but are first subject to quantization (Gray & Neuhoff, 1998). This involves reducing the bitwidths below typical floating-point precision. Following quantization, the parameters are processed through entropy coding, inour experiments we utilize Brotli coding (Jones & Jones, 2012; Alakuijala et al., 2018), which allows the compressed data to be stored or transmitted efficiently. To retrieve the original parameters, the decoder must reverse the entropy coding and then perform dequantization. In the case of **CoINR**, the compression process is intensified by utilizing the sparsity induced in the model parameters by natural signals. Once the sparse code is established, the parameters are quantized and subjected to entropy coding. The decoder then reverses the entropy coding and dequantizes the data. Finally, the model parameters are reconstructed by multiplying them with a random Gaussian matrix, which is determined by a specific seed.

## 4 EXPERIMENTS

### 4.1 EXPERIMENTAL SETUP

**CoINR**, a novel INR compression algorithm, is predicated on the idea that if natural signals are compressible through a dictionary, then INRs should be similarly compressible. This concept underpins **CoINR**'s goal to efficiently reduce INR storage requirements while maintaining high fidelity. Our experiments, conducted using the PyTorch framework following WIRE (Saragadam et al., 2023) codebase on an NVIDIA RTX A5000 GPU with 24 GB of memory, spanned various data types

including images, occupancy fields, audio, and neural radiance fields. Image encoding metrics involved file size, bits per pixel (bpp) and Peak Signal-to-Noise Ratio (PSNR). Occupancy fields were evaluated using file size and Intersection over Union (IoU), and neural radiance fields were assessed using file size and PSNR. Other than the network configurations mentioned in the paper, for occupancy field evaluation, we utilized an MLP with 128 hidden neurons, and 3 hidden layers. For INRIC, we applied the network hyperparameters specified in its paper. In COIN++, we followed the guidelines in its paper but modified the hidden neuron size to 300. All experiments used Brotli entropy coding with a 16-bitwidth (65536 levels) uniform quantizer.

## 4.2 How do we find s ?

We implemented $L_1$ minimization using the Orthogonal Matching Pursuit (OMP) algorithm (Tropp & Gilbert, 2007). The OMP algorithm requires the pre-determination of $s$ before obtaining $\mathbf{x}$, and it must adhere to the condition $2s < k_1$. If $2s$ is set too low, it results in inaccurate representations of $\mathbf{w}$ within the weight space. Therefore, we incrementally increased $s$ from a low value until $2s = k_1$ for all KODAK images in the $C_1$ experiment, as outlined in section 4.3. Our findings suggest that the optimal value of $s$ for successfully reconstructing the weight space does not depend on the specific image but on the number of neurons in a hidden layer. By adjusting the neuron count, we identified an optimal $s$ that accurately reconstructs the weight space while satisfying the specified constraint. Extending these experiments to natural signals outside the KODAK dataset confirmed the consistency of our results. Consequently, we have included a regression plot in the supplementary material that details how to determine the optimal $s$ based on the number of neurons.

## 4.3 Image encoding

Representing an image through the weights and biases of a neural network serves as a method of encoding. For our image encoding task, we utilized the KODAK dataset, which includes 24 natural RGB images, each measuring $768 \times 512$ pixels. We conducted five types of experiments, denoted as $C_i$, where $i$ ranges from 1 to 5, to demonstrate the effectiveness of our proposed method.

Experiment $C_1$ involved encoding each image in the KODAK dataset using an INR without positional embedding, by varying the number of neurons in each hidden layer. Experiment $C_2$ mirrored $C_1$, but with the variation in the number of hidden layers instead. Experiments $C_3$ and $C_4$ implemented the meta-learning approach for INRs proposed in INRIC, without and with positional embedding for the input layer, respectively. For these meta-learning-based experiments, we used the first 12 images of the KODAK dataset for meta-learning and the remaining 12 images for fine-tuning.

Experiment $C_5$ involved the COIN++ framework, testing both with and without patching. When using patching, we adopted $32 \times 32$ patches as suggested by COIN++. However, we observed that without patching, even as the latent modulation dimension increased, the average Peak Signal-to-Noise Ratio (PSNR) obtained by COIN++ remained nearly constant. For all image encoding experiments, we used the sinusoidal activation function (see supplementary).

Let us define $h$ and $m$ as the number of hidden layers and the number of neurons per hidden layer in an INR, respectively. For experiment $C_1$, we configured the INR with settings $(h, m)$ as $(2, 32), (3, 64), (3, 128)$. Experiment $C_1$ aims to assess the effectiveness of **CoINR** by varying the number of hidden neurons. The results, depicted in figure 3, demonstrate how effectively **CoINR** identifies the compressibility of the weight space. This is indicated by the bits-per-pixel (bpp) values, which reflect the size of the model parameters. For example, representing the KODAK dataset with an average PSNR of 30 dB requires about 3.7 bpp for COIN and 2.0 bpp for INRIC. However, **CoINR** significantly reduces the bpp to approximately 1.7 using the same quantizer and entropy coder. The first configuration in $C_1$ falls under the category of tiny INRs, underscoring the proposed method's effectiveness even for compact INRs. As illustrated in figure 3, **CoINR** achieves the same level of PSNR as baselines with a lower bpp for any network configuration. This substantial reduction of bpp across the $C_1$ experiment showcases the efficiency and compactness achieved by **CoINR**. From $C_1$, it can be established that greater compressibility of an INR into a dictionary is possible with an increased number of hidden neurons. Following the conclusions drawn from experiment $C_1$, experiment $C_2$ was designed to explore the impact of increasing the number of hidden layers on the

effectiveness of **CoINR**. The configurations tested in $C_2$ were $(h, m) = \{(3, 64), (5, 64), (7, 64)\}$. As illustrated in figure 3, **CoINR** consistently achieved PSNR levels comparable to baseline methods, but with a reduced bpp. Given that $C_2$ maintained a constant neuron count at 64, the observed deviations in compression between **CoINR** and INRIC were less significant than those observed in $C_1$. This discrepancy can be attributed to the following: an INR configuration with a higher number of neurons (e.g., $m = 128$), even with fewer hidden layers (e.g., $h = 2$), possesses more trainable parameters. Consequently, such a model is capable of learning a more robust representation of the image compared to configurations with a larger number of layers but fewer neurons per layer. As a result, the compressible characteristics of the images are more effectively transferred into the model parameters during the INR training process. This leads to a more compressible INR. These findings support the premise that if natural images can be efficiently compressed into a dictionary, the weight space of INRs can also be effectively compressed.

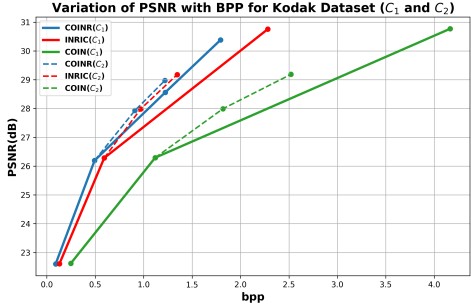

Figure 3: **Experiments $C_1$ and $C_2$: Identifying compressible INR combinations**. The **CoINR** approach demonstrates that configurations in $C_1$ are more compressible than those in $C_2$. Furthermore, in both configurations **CoINR** achieves lower bpp while maintaining the PSNR values.

As in previous experiments, each image required separate training of an INR. Experiments $C_3$ and $C_4$ address this challenge through meta-learning, with and without positional embedding, respectively. The configuration for these INRs is given by $(h, m) = \{(3, 32), (3, 64), (3, 96), (3, 128)\}$. For COIN++, the number of layers was set to 5 with MLP's hidden dimension at 300. The latent dimension parameter ($d$) varied as follows: $d = \{16, 32, 64, 96\}$. Figure 4 presents the experimental results for $C_3$, $C_4$, and $C_5$, illustrating significant compression capabilities of the proposed **CoINR** within a meta-learning framework. Notably, models using positional embedding generally have more parameters than those without.

Comparing the performance of INRIC and **CoINR** without positional embedding schemes, the initial INR configuration shows that **CoINR** exhibits a lower bpp for the same average PSNR. Generally, as bpp increases, the representation capacity of the INR enhances, leading to more robust image representation. At higher bpp values, the **CoINR** graphs demonstrate a greater deviation from the INRIC graphs, a phenomenon that can be explained by the aforementioned logic.

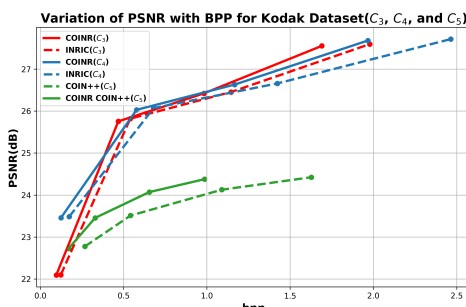

Figure 4: **Experiments $C_3$, $C_4$, and $C_5$: Identifying compressible INR combinations mnder Meta-Learning**. Meta-learning approaches have been introduced for INRs to enhance their generalization abilities and achieve faster convergence. When assessing induced sparsity in the weight space, **CoINR** demonstrates a significant reduction in bpp values while maintaining nearly the same PSNR performance as the baselines.

In the case of COIN++, the approach focuses on fine-tuning only the modulations using their proposed meta-learning method. However, since fine-tuning encodes natural signals within these modulations, they should be compressible via a dictionary. Due to patching, each KODAK test image results in a $d \times 384$ matrix. Our experiments reveal that these modulations encode hidden redundancies in natural signals. For instance, to achieve an average PSNR of approximately 24.2 dB, COIN++ requires more than 1.5 bpp; however, the same PSNR can be achieved with COIN++ using just under 1 bpp by exploiting the hidden sparsity in its modulations through our proposed approach. Therefore, when a high-capacity model effectively represents a signal, it must encapsulate this sparsity within its weight and bias spaces. **CoINR** explores and removes redundancies in these parameters, retaining only essential information. Figure 5 showcases the decoded images by **CoINR** alongside

with the INR based image compressors. Decoded PSNR, BPP, and file size are displayed in the first, second, and third rows of the text boxes

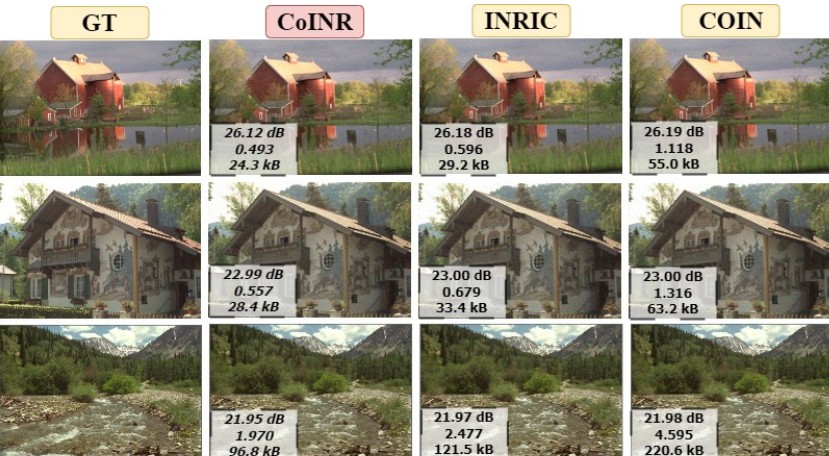

Figure 5: **Results for image encoding experiment**. `CoINR` compresses the INR into a dictionary, significantly reducing the storage required compared to baseline INR image compressors. The results demonstrate that the decoded representations undergo a very negligible loss in PSNR, which is minimal considering the substantial storage space saved.

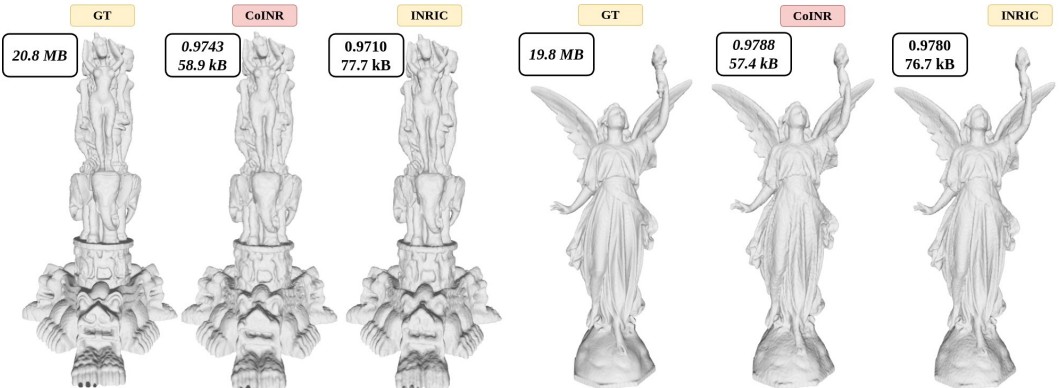

Figure 6: **Results for occupancy fields encoding experiment**. The results clearly demonstrate that `CoINR` achieves the smallest file size and the highest accuracy metric for every shape in the tested dataset. The significant compression obtained by our algorithm suggests that occupancy fields, when represented using an INR, can be more efficiently compressed into a dictionary compared to images. This may be attributed due to the inherent redundancies present in the occupancy fields.

## 4.4 OCCUPANCY FIELDS ENCODING

Occupancy fields are represented by binary values, either 1 or 0, where 1 denotes that the signal lies within a specified region and 0 indicates its absence. Another variant of occupancy volumes stores not only the presence or absence of a signal but also the color at that location. Typically, occupancy fields consume more space than other data modalities. However, they can be represented with higher accuracy and lower storage requirements using INRs. In this experiment, we followed the sampling procedure described in Saragadam et al. (2023). Occupancy fields can be thought of as representations of three-dimensional objects, capturing natural signals. Despite following the sampling procedure, redundancies may exist that are not essential for representing the occupancy volume. Identifying these redundancies can reduce storage requirements. However, identifying them in the spatial domain (xyz) requires domain-specific algorithms, as described in section 1.

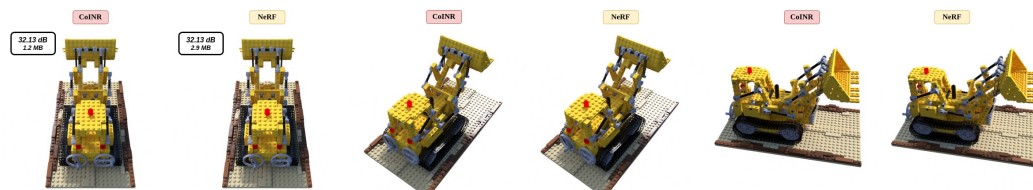

Figure 7: **Results for NeRF compression: `CoINR`** compresses the radiance field without any loss in PSNR while significantly reducing storage requirements.

As INRs serve as unified data modality representators, these redundancies must be encapsulated within its weights space. **CoINR** fundamentally compresses the INRs into a dictionary regardless of the data modality; therefore, indeed it is equally applicable to occupancy fields. To validate this hypothesis for occupancy fields, two experiments were conducted using shapes from the Stanford shape dataset (Stanford University Computer Graphics Laboratory). Figure 6 showcases the decoded **CoINR**'s representations for 'Thai Statue'(first volume) and 'Lucy' (fourth volume) datasets alongside the existing INR-based occupancy compressor. We use the Gaussian activation function for this task (see supplementary). The first value and second value in each text box represent the IoU metric and storage requirement, respectively, except for GT.

### 4.5 NEURAL RADIANCE FIELDS ENCODING

NeRF can be considered a novel view generator when it is trained with a sufficient number of training views, along with their corresponding positions and directions. Fundamentally, once trained, a NeRF is an INR. Therefore, the information encoded in its weights for generating novel views can be compressed into a dictionary. Figure 7 presents the results obtained with the proposed **CoINR**. As shown, **CoINR** achieves more than 50% compression while maintaining the same PSNR. These results further confirm the applicability of **CoINR** for compressing INRs across different data modalities. We used the ReLU-PE activation for encoding NeRFs.

### 4.6 ADDITIONAL MATERIALS

The pseudocode for **CoINR**, additional results, and ablation studies on finding $s$ are available in the supplementary material.

## 5 CONCLUSION

Implicit Neural Representations (INRs) have emerged as a promising framework for unified data modality representation. Several studies have explored the potential for compressing images, occupancy fields, and audio using INRs. However, none of these methods have investigated whether the INR itself can be compressed prior to quantization and entropy coding. As natural signals can be efficiently compressed in bases of transformed domains due to their sparsity—allowing for higher accuracy and lower storage requirements—we hypothesize that a similar compressible nature must also exist in the INR once it is trained. With the discovery that weight vectors in the weight space tend to adhere to a Gaussian distribution, we propose **CoINR**, which compresses any INR in a dictionary. Furthermore, we demonstrate that this dictionary does not need to be learned but can instead be generated using a seed. We compare our findings with standard INR compressors for images, occupancy fields, and neural radiance fields. **CoINR** achieves fundamental compression for any INR, independent of other post-processing methods such as quantization and entropy coding, and it showcases significantly lower storage requirements and higher fidelity across various data modalities. Through our experiments, we observed that the INR can be more compressed when a more robust representation of the signal is learned. Additionally, some data modalities exhibit greater compressibility than others. We firmly believe this research will aid other researchers in exploring more patterns in the weight spaces of INRs and in developing operators and transforms for INR.

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
