# OpenReview forum: "CoINR: Compressed Implicit Neural Representations"
_ICLR.cc/2025/Conference — ICLR 2025 Conference Withdrawn Submission_

### Official Review · Reviewer_1Ggt · 2024-10-21

**Soundness:** 2
**Presentation:** 2
**Contribution:** 2
**Rating:** 5
**Confidence:** 4

**Summary:**

This paper proposes to encode INR by transmitting the weight into a sparse code through a randomly initialized matrix A.
Experiments on images, occupancy fields, and NeRF showcase the effectiveness of this method.

**Strengths:**

1. This paper exploits the sparse structure in INR weights to encode INRs more efficiently. Across various datasets and modalities, this method demonstrates significant improvements over existing baselines.
2. The approach is plug-and-play. It does not rely on a specific structure of INRs and can be applied to many estimated approaches.
3. I found the analysis of sparsity levels for different hidden neurons in the appendix particularly interesting. It reveals that the number of dense elements scales roughly linearly with the number of hidden neurons.

**Weaknesses:**

1. There are many missing related works and baselines. For NTC, [1] and [2] are two of the most influential works that should be mentioned in intro or background; for INRs, there are more recent works after COIN++ and INRIC. [3] encode LoRA-style modulation with a learned VAE; [4] investigates the Gaussian pattern in INR-weights and encode a Bayesian INR; [4] further exploit this idea, by introducing a (learned) linear transformation to the INR weights.
All of these works exploit the sparse pattern in INRs via different ways.
The approach proposed in this paper differs from these works in that the transformation is random and does not require learning. However, these are closely related works that need to be mentioned and ideally compared.

2. Because of these approaches, I think it is not true to claim 'None of the existing methods, however, have explored fundamentally compressing the INR by identifying patterns within its parameter space before applying standard techniques such as quantization and entropy coding'.

3. The experiments for COIN++ are a bit confusing. The author discussed that their approach can also reduce the compression rate for the base network. However, COIN++ does not need to encode the base net.
This follows the same argument as the VAE-based approach: some common structures and parameters can be shared without transmitting.
Also, it seems that the curve for COIN++ in Fig 4 is much lower than reported in the COIN++ paper. Is this because you take the base network into account?

Related to 3. How does COIN++ (combined with the proposed approach) perform when not considering the base network? Will this approach still help?

4. The result for NeRF appears somewhat unfair—correct me if I'm wrong.
The authors compared the file size of NeRF with their proposed approach.
However, as far as I understand, NeRF's file size is not compressed, which makes the comparison less fair.
A more appropriate comparison would involve NeRF with quantization (to a slightly lower precision) and entropy coding applied.


5. As the weight matrix A is randomly initialized, I would expect this approach to exhibit a certain level of randomness. Is this true? Seeing the performance's error bar across several different seeds would be nice.


I will be happy to raise my rating if my concerns are well addressed.


[1] Ballé, Johannes, Valero Laparra, and Eero P. Simoncelli. End-to-end optimized image compression. ICLR 2017.

[2] Ballé, Johannes, et al. Variational image compression with a scale hyperprior. ICLR 2018.

[3] Schwarz, Jonathan Richard, et al. Modality-Agnostic Variational Compression of Implicit Neural Representations. ICML 2023.

[4] Guo, Zongyu, et al. Compression with bayesian implicit neural representations. NeurIPS 2023.

[5] He, Jiajun, et al. RECOMBINER: Robust and Enhanced Compression with Bayesian Implicit Neural Representations. ICLR 2024.

**Questions:**

1. The authors employed different strategies for larger and smaller INRs. I would expect that vectorizing the weights in larger INRs (the approach for tiny INRs) could also be beneficial, as it might better capture complex dependencies across the entire weight matrix. Since the matrix A is not learned, using a relatively large matrix A should be acceptable. Is this a reasonable assumption? Will this provide more gains than what we see in the paper now?


2. Line 343-344: "Extending these experiments to natural signals outside the KODAK dataset confirmed the
consistency of our results." Do you mean the hyperparameters tuned on Kodak can be applied to other datasets? Or do you mean the same hyperparameter tuning strategy also works for other modalities?

---

### Official Review · Reviewer_c2ZP · 2024-11-01

**Soundness:** 2
**Presentation:** 2
**Contribution:** 2
**Rating:** 5
**Confidence:** 3

**Summary:**

The paper addresses the issue of implicit neural representation (INR) compression. The authors use a sparse code and dictionary approach based on compressed sensing principles to improve compression beyond existing quantization and latent code approaches. The decoder sensing matrix can treated as a random matrix, enabling the weights to be reconstructed by transferring the sparse code + a random seed. The paper shows that this can lead to improved rate-distortion performance for images and occupancy field, and can be used to compress neural radiance fields. While the method is well-motivated and the results show promise, there are a number of important issues of clarity and presentation that would help to improve the quality of the paper.

**Strengths:**

- The topic of implicit neural representation compression is important, especially given the number of applications and large memory requirements of standard approaches. Improvements to post-processing methods for INR compression is an area that would could have broad applicability.

- The compressed sensing / dictionary approach described by the paper is described clearly and motivated well. The use of a random sensing matrix is well motivated and is a nice way to minimise the data transfer.

- The method is demonstrated on a range of implicit neural experiments including images, occupancy fields, and neural radiance fields. Improved performance is shown relative to baselines (INRIC and COIN).

- The paper is broadly written clearly, although some aspects could be re-ordered, clarified or explained in more detail.  Figure 2 is nice and gives a great summary of the method.

**Weaknesses:**

The main issues of the paper relate to clarity and presentation.

Major Presentation:
- L205-225 lists alternative methods tried for the paper, before describing the dictionary method used. It may be clearer listing these in the discussion or limitations instead (the paper order doesn't need to reflect the order that methods were tried). A few methods discussed (e.g. structured / unstructured pruning) don't appear to be used in the main paper method / experimental comparisons.

- It may be useful to list experiments C1-5 either in bold text or in sub-sub sections. Section 4.3 and Figures 3, 4 are difficult to parse and require looking at different areas to understand the experimental details / architectures. These figures largely appear to involve ablations (e.g. with / without positional encoding, different architectures), so reorganisation may improve focus on the key findings and improve readability.

Related Literature:
One area which could be improved is the discussion of related literature. Compression of implicit neural representations is a rapidly growing field, and there are a number related approaches. While experimentally comparing to all methods may not be necessary (the COIN / INRIC baselines appear appropriate), placing CoINR in the context of other INR compression methods would be useful. For example (non-exhaustive):
- Dictionary-based INR approaches: Yüce et al. (2021) https://arxiv.org/abs/2112.01917
- Sparse INRs: Lee et al. (2021) https://arxiv.org/abs/2110.14678
- Occupancy Field Compression: Lu et al. 2021 https://arxiv.org/abs/2104.04523
- NeRF compression: e.g. Bird et al. (2021) https://arxiv.org/abs/2104.12456, Isik et al. (2021) https://arxiv.org/abs/2105.03120, Shi et al. (2023) https://arxiv.org/abs/2208.00164, Takikawa et al. (2023) https://arxiv.org/pdf/2312.17241, etc

Experiments:
- Compression results for NeRF are only compared to the baseline NeRF. It would be useful to experimentally compare to a direct quantization + entropy compression baseline (similar to the COIN / INRIC comparisons for images and occupancy fields).
- Figures 12-15 (Supplementary) it is not clear why COIN results are not included (as in Figures 10 and 11)

Minor Presentation:
- The text in Figures 1, 3, 4, 6, 7, 8, 9, 10, and 11 is too small to be easily read.
- Qualitative differences are difficult to see when zooming in (it may be useful to use a zoomed in 'cut out' to make the improvements clearer).

**Questions:**

- Figure 2 + Algorithm 1 indicate that the method generates a sparse code directly from a pre-trained INR. While it's assumed that the method is a post-processing step, this isn't explicitly mentioned in the main text. Could the authors clarify whether any additional training / fine-tuning is required? Any details about the algorithm's optimization process (L726), or run time would be very useful.

- Could the authors clarify whether the method can be applied to different pre-trained INRs, and does this lead to different results (e.g. for Figures 10 / 11 is the method applied to INRs trained with INRIC, COIN, or a different base INR)

General Suggestions:
- Analysis of compression results can be improved by including BD-Rate / BD-PSNR figures https://arxiv.org/abs/2401.04039
- Different activation functions are used in experiments based on ablations in the Supplementary Materials. While this demonstrates the method is flexible to work with different activations, it would be useful to discuss the hyperparameters used for the Gaussian / SIREN activations (as performance depends often critically on these values).

---

### Official Review · Reviewer_q3Mz · 2024-11-03

**Soundness:** 3
**Presentation:** 2
**Contribution:** 2
**Rating:** 3
**Confidence:** 4

**Summary:**

This paper proposes to use compressed sensing for the task of data compression with implicit neural representations(INRs). First, the authors reveal by experiments that the neuron weights of INR follow the Gaussian distribution. Therefore, according to the Central Limit Theorem, a sparse code x can be optimized based on a random sensing matrix A. By transmitting the sparse code x, the codec does not need to transmit the whole high-dimensional neural weights. Experiments demonstrate this method can outperform the COIN++ and INRIC on tasks including image compression, occupancy field compression and neural radiance field compression.

**Strengths:**

The idea of using compressed sensing in INR compression makes sense, as early INR compression methods, such as COIN, suffer from the redundant information among neuron weights.

**Weaknesses:**

1. The core idea of this paper makes sense but is more like a simple combination of existing INR compression and well-defined compressed sensing (SC) technique.

 2. And the performance of INR+SC is actually not good enough if compared with other recent INR compression methods. I don’t think this idea is very promising in the future that can be followed and developed continuously.

 3. There are a few more detailed problems in this paper, including

(a) incomplete literature review for INR compression, such as missing references to
[Ref1] Compression with Bayesian Implicit Neural Representations. Guo et al., NeurIPS 2023.
[Ref2] Modality-agnostic variational compression of implicit neural representations. Schwarz et al., ICML 2023.
COIN++ and INRIC were published two years ago. The abovementioned new INR compression works (not limited to the above two) should also serve as baselines for comparison.

(b) although the compressed sensing seems to help the performance at high bitrate in image compression. Actually the performance still has a huge gap compared with both traditional codecs like BPG or variational image compression models such as [Ref3].
[Ref3] Variational Image Compression with a Scale Hyperprior. Ballé et al., ICLR 2018.

(c) For occupancy field encoding and neural radiance field encoding, it would be better to include RD curve comparisons.

**Questions:**

See in the Weakness section.

**Details Of Ethics Concerns:**

No ethics concerns.

---

### Official Review · Reviewer_MvLm · 2024-11-03

**Soundness:** 1
**Presentation:** 1
**Contribution:** 2
**Rating:** 1
**Confidence:** 3

**Summary:**

The authors propose a post hoc method for compressing neural network parameters; specifically, they consider compressing the weights and biases of implicit neural representations trained on various data modalities.

Concretely, given the weights represented as a vector $w \in \mathbb{R}^n$ of a trained neural network, the authors propose to reduce the parameter count using the following sparse weight-sharing technique:
 1. They randomly sample a matrix $A \in \mathbb{R}^{n, m}$ with $n \ll m$.
 2. They find a sparse representation $x$ by solving $\min\lVert x\rVert_1$ subject to $w = Ax$.
 3. They use a variant of ZIP called Brotli to compress the indices and values of the non-zero entries of $x$.

The authors test their method on various data modalities and demonstrate that it improves the rate-distortion performance of well-known INR-based compression methods such as COIN, COIN++ and INRIC.

**Strengths:**

The authors developed a post hoc method for compressing model parameters, which can be used to improve other compression techniques. Furthermore, the sparse coding idea is neat, and I have not seen it explored elsewhere.

**Weaknesses:**

The work suffers from severe weaknesses in its analysis and presentation and contains incorrect mathematical statements.

To begin with, the two grossest offenders:
 1. On line 238, the authors claim that "According to the Central Limit Theorem (CLT), a normally distributed random variable can be produced through a finite linear combination of any random variables" - this statement makes multiple incorrect assertions. The CLT does not guarantee Gaussianity in a non-asymptotic regime and most certainly does not hold for a finite linear combination of arbitrary random variables.
 2. On the same line (L238), the authors claim, "As we have confirmed, the weights are normally distributed". As far as I can tell, this statement is not even approximately correct from any sensible perspective. The assertion that the marginal weight distribution is Gaussian is disproved by the authors' own Figure 1b, which depict marginal weight distributions that are either significantly more concentrated (Fig 1b, left) or significantly more heavy-tailed (Fig 1b, right) than a Gaussian. In fact, it is not even clear what data the authors are plotting: are they plotting histograms of the weight distribution of a single INR (I believe this to be the case), or are they plotting the weight distribution by collecting the weights from the same INR architecture trained on different data points?

Unfortunately, based on these two points, I believe the paper is already potentially beyond repairability to be published at ICLR. However, should the authors wish to improve their paper to be publishable, beyond correcting the above issues, there are further matters that need attention:

 - Ablation studies are missing: how does quantization affect reconstruction quality? How do different distributions for the entries of the sensing matrix impact the results?
 - Missing related works: the authors should discuss the works of [1] and especially [2], as the latter includes a similar linear weight-sharing technique as the authors propose here.
 - Clarifying the use of the term entropy coding: The authors use Brotli coding  - a universal source coding method based on the Lempel-Ziv algorithm. Claiming that this is entropy coding is somewhat misleading, as in the literature, entropy coding usually implies that we build an explicit statistical model and use it in conjunction with arithmetic coding to encode the data.
 - Most figures in the main text are illegible at 100% zoom. Please increase the font size of labels, legends and tick marks to match the caption font size.
 - While I appreciated the explanation of the attempted but failed compression techniques the authors tried in Section 3.2, they should provide experimental evidence, as the discussion is currently based on anecdotal evidence only.
 - More minor: The first paragraph of the introduction is very general and can be safely removed.

## References
- [1] Guo, Z., Flamich, G., He, J., Chen, Z., and Hernández-Lobato, J. M. (2023). Compression with Bayesian implicit neural representations. In NeurIPS 2023
- [2] He, J., Flamich, G., Guo, Z., and Hernández-Lobato, J. M. (2024). Recombiner: Robust and enhanced compression with bayesian implicit neural representations. In ICLR 2024

**Questions:**

n/a

---

### Note · Authors · 2024-11-15

I have read and agree with the venue's withdrawal policy on behalf of myself and my co-authors.